# The Effect of Improving Cycleway Environment on the Recreational Benefits of Bicycle Tourism

**DOI:** 10.3390/ijerph16183460

**Published:** 2019-09-17

**Authors:** Chun-Chu Yeh, Crystal Jia-Yi Lin, James Po-Hsun Hsiao, Chin-Huang Huang

**Affiliations:** 1Department of Tourism and Hospitality, Transworld University, Yunlin 640, Taiwan; joyceyehh@gmail.com; 2Department of Physical Education, National Taiwan University of Sport, Taichung 404, Taiwan; lincrystal@ntupes.edu.tw; 3Department of Sport Management, National Taiwan University of Sport, Taichung 404, Taiwan; phhsiao@ntupes.edu.tw

**Keywords:** bicycle tourism, environment quality, recreational benefits, contingent behavior method

## Abstract

Bicycle tourism is one of the popular physical activities for sport tourists. Since the physical environment may affect bicycling behavior, it becomes an important determinant for cyclists to choose a cycleway. Exploratory factor analysis is performed to extract the perception of environmental quality of cyclists into five main factors, including safety, light facilities, lane design, landscape, and environment cleanliness. The contingent behavior method (CBM) is adopted to measure the quality improvement projects in different scenarios of light facility and landscape improvement. The results showed that the improvement projects increased the intended number of trips and the recreational benefits of cyclists.

## 1. Introduction

Bicycle tourism has an important niche in the tourism market, and it is defined as ‘tourism that involves watching or participating in a cycling event, or participating in independent or organized cycle touring’ [1]. Taiwan’s bicycle industry is famous for the bicycle parts it produces and the bicycles it assembles. Due to Taiwan’s natural environment, bicycle routes in Taiwan are unique and unprecedented. For example, the course of the Taiwan KOM (King of the Mountain) Challenge climbs from 0 to 3275 meters above sea level for a total route length of about 105 kilometers. Participants of Taiwan KOM are challenged by steep slopes and enjoy natural scenery. The environmental quality of KOM’s bicycle routes includes smooth roads, climbing sections, and beautiful views of mountains which attract many international cyclists to the event. The environmental requirements for bicycle tourism are rather different from those of city sightseeing and festival forms of tourism [2]. Cyclists need a special environment for cycleways [3]. An attractive environment can appeal to racing cyclists [4]. They also strongly prefer scenery, cycling routes, and quiet roads [5]. The physical environment is an important determinant of consumers’ perceptions of chosen destinations [6]. However, there has been little well-developed research on the environmental impacts of off-road cycling and there are few quantitative studies on the impacts of mountain bike trails [7]. There is a lack of research estimating the monetary value of environment quality in terms of cycleways. This study estimates the effect of environment quality on bike tourism.

Through cycling events, bicycle tourism can bring economic, social, and environmental impacts to the host communities and individual participants [2]. Studies on bicycle tourism often focus on motivations [8,9], general characteristics [10], bicycle road racing subculture [11], and gender differences [12,13] of competitive cyclists. Previous studies rarely look at the environment quality that is needed for bicycle tourism [3], and it is difficult to evaluate the quality of the environment at a recreation site [14]. Since the effect of environmental quality and recreation benefit cannot be estimated by market price, they are considered nonmarket goods. This study adopted the contingent behavior method to estimate the effect of environmental quality improvement. 

According to the Ministry of Transportation and Communications in Taiwan (2018) [15], the number of cyclists increased rapidly from 700,000 in 2008 to 2.45 million in 2013, and in 2017 the number of cyclists increased to 5.1 million. Around 80% of these cyclists cycle for recreational purposes, resulting in a greater demand for dedicated bicycle routes [16]. Most cyclists in Taiwan cycle for leisure or recreational bike tourism. Dong-Feng Cycleway, also known as the Green Corridor, is one of the most popular cycleways in central Taiwan. It was built along an abandoned railway, connecting Fengyuan and Dengshi districts in Taichung City. The cycleway stretches for 12 kilometers with a river on one side and trees on both sides of the path, offering a great view to cyclists. Figure 1 shows a diagram of the site and pictures of Dong-Feng Cycleway. The built environment of Dong-Feng Cycleway is safe and comfortable and attracts many cyclists. This study focused on leisure bike tourism, and the purpose of this study was to explore the influence of environmental quality on the demand for cycleways, and to estimate the effect of environmental improvement on recreational benefits.

## 2. Literature Review.

Bicycling is recognized as a sustainable travel mode and an important form of physical activity [17]. Bicycle tourism can be defined as ‘tourism that involves watching or participating in a cycling event, or participating in independent or organized cycle touring’ [1]. Lamont (2009) has expanded the definition to ‘the scope for investigating the relationship between cycling and tourism by justifying the inclusion of persons who travel for the purpose of engaging in competitive cycling, in addition to persons who travel specifically to observe cycling events’ [18].

Social and environmental factors affect cycling choice behavior, including demographic, environmental, and geographic variables [19]. For example, people’s perceptions of the environment—their awareness of the recreation site through their primary receptive senses—can have a direct and significant influence on bicycling behavior. In contrast, the objective environment may only affect bicycling behavior indirectly through influencing cyclists’ perceptions [20]. Previous studies have identified that cyclists’ preference can be affected by the environment and bicycle facilities [21,22].

The main factors affecting recreational cyclists’ choices include bicycle route choice, basic bicycle facilities, bicycle lane type, roadway grade, and scenery. Cycling routes can be divided into two types, commuting and recreational routes [21]. This study focused on recreational cycling. In Taiwan, recreational cyclists with higher skill levels prefer challenging routes and varied bicycle touring experiences. Cyclists also prefer cycling routes that are near attractions, cycling facilities, information centers, and bike-specific paths [16]. Road surface quality in particular is one important determinant in destination attractiveness [23]. With regard to safety on the road, bicycle lanes, bicycle slots, and wide curb lane are important factors; other factors include clean and smooth roads, route safety, diverse scenery, length of ride, and route variety. Creating bicycle infrastructure can induce more bicycling, and can influence cyclists’ decisions to take on cycling touring [24,25]. Factors such as beautiful scenery or countryside were also reported to have a strong influence on sport tourism and customer satisfaction [26]. Based on previous research, cyclists’ perception of the cycleway’s environmental quality is rather important. 

Perceived environmental quality can influence tourists’ decisions [27]. Omitting the effect of the environmental quality from a demand model would result in underestimation of recreational benefits and lead to poor decision-making [28]. Therefore, to improve participants’ perceptions of the environment is as important as it is to improve the physical environment and cycling infrastructure, and should be seen as a way to complement the design of the built environment [20]. Since the environmental quality is the main factor determining the behavior of participants [29], and the effect of environment belongs to nonmarket value, scholars have applied contingent valuation (CVM) to estimate their willingness to pay (WTP) as a monetary value. However, CVM described a hypothetical scenario that incurs hypothetical bias and the hypothetical WTP differs from actual WTP [30]. In order to mitigate the hypothetical bias of CVM, Whitehead and Wicker (2018) performed willingness to travel (WTT) to revise the hypothetical bias of WTP [30]. They combined stated and revealed preference data and asked respondents their intention of revisiting alternative distance projects for cycling events. The WTT is similar to the contingent behavior method (CBM) that Whitehead et al. (2000) had suggested to estimate the recreation benefits for the improvement of environmental quality [14]. Yeh, Hua, and Huang (2016) performed CBM to evaluate the improvement value of service quality for sports tourism [31]. Huang (2017) adopted CBM that combined actual and intended behavior data to measure the environmental effects of quality improvement [28]. Deely et al. (2019) combined actual and contingent behavior data to estimate the value of coarse fishing in Ireland [32]. This study also adopted CBM to estimate the improvement effect of environmental quality for cycleways. 

## 3. Materials and Methods

### 3.1. Contingent Behavior Method

The environmental quality of recreation sites has been included in demand functions to estimate consumers’ willingness to pay [33]. However, it is difficult to estimate the environment quality at the same recreation site due to there being no variation in quality data [14]. The problem is how to evaluate the improvement effect and to identify the changes in quality variation that are associated with recreational benefits [14,34]. 

The most common approach to evaluate the quality improvement effect is to combine revealed and stated data, the so-called contingent behavior method [14,35,36,37,38,39,40]. This means a panel recreation demand model combining current data and expected hypothetical scenarios is used to measure consumer benefits under different projects [36]. This study also adopted CBM to estimate the improvement effect of environmental quality for cycleways.

The estimation model of this study was based on the travel cost method. Then, the questionnaire was designed to ask respondents about their observed behavior from actual trips and their intended behavior with hypothetical changes under certain circumstances, such as improved environmental quality. The contingent behavior question asked subjects whether they would increase the number of their visits if the environmental quality of Dong-Feng cycleway were improved. Then, actual and intended data were combined to create a panel data set that was generated from one cross-sectional sample survey. The advantage of combined data is its efficiency and reduction of sample sizes from repeated observations for each individual without incurring additional costs. The recreational benefits can be measured by the change in consumer surplus between the demand function of actual trips and intended behavior trips.

This study followed previous research using CBM to estimate recreational benefits [28,31,32,35,37,39,41]. The CBM combined actual trips with contingent behavior data regarding visit intentions given alternative projects. Panel data of the recreation demand model with pooled data of current and expected hypothetical scenarios was applied to measure consumer benefits under different projects [36]. The random effects Poisson model was employed to take into account the heterogeneity among individuals and structural changes in demand in different scenarios [31,42,43]. The Poisson probability density function is as follows:(1)$$P(X_{it}=x_{it})=\frac{e^{-\mu_{ti}}\mu_{it}^{x_{it}}}{x_{it}!},\;x_{it}=0,1,2,\ldots$$

Assume $x_{it}$ is the number of times taken by individual i in a scenario t, and $\mu_{it}$ is the mean Poisson distribution, which depends on the explanatory variables and participant heterogeneity:(2)$$\ln u_{it}=\alpha_{t}+\beta_{t}COST_{it}+\delta_{t}SCOST_{it}+\varphi_{t}INCOME_{it}+\gamma_{t}OTHER_{it}+u_{i}$$
where t = 1, 2 and $u_{i}$ is a random effect for respondents i. Where t = 1 indicates the current level of lighting facilities and landscape and t = 2 represents the improvement scenario of lighting facility and landscape. COST represents respondents’ travel costs, including immediate transportation costs and the cost of round-trip travel time from their home to the destination, as well as time spent on-site. SCOST represents the travel cost associated with a visit to a substitute site. The substitution price is measured by the distance from the home of a visitor to an alternative site that offers similar attractions and includes the same expenditure as the site under study. Respondents were asked where they would go to make a trip if they did not go to Dong-Feng Cycleway (the Green Corridor). The most frequent choice of the respondents for the substitute site is Kenting National Park in south Taiwan. INCOME is monthly income of the respondents. OTHERS includes the main factors of environmental quality and AGE. In order to account for the potential structural change in trip demand across scenarios, this study combined data from all trip scenarios. The dummy variable D = 1(t = 2), denotes the improvement programs of lighting facilities and landscape; otherwise, D = 0 (t = 1). The definition of the variables and descriptive statistics are listed in Table 1. A general recreation demand model uses pool data to incorporate the dummy variable into the mean $\mu_{it}$.
(3)$$\begin{array}{l} \ln\mu_{it}=\alpha_{t}+\beta_{t}COST_{it}+\delta_{t}SCOST_{it}+\phi_{t}INCOME_{it}+\gamma_{t}OTHERS_{it}\\ +a_{2}D_{s}+b_{2}D_{s}COST_{it}+c_{2}D_{s}SCOST_{it}+d_{2}D_{s}INCOME_{it}+u_{i} \end{array}$$
where D_s_ represents the dummy variable for improving programs, s = 1, 2. When the coefficient of the dummy variable is significantly different to 0, it means that the visitors’ motivation to ride a bike will be raised after the lighting facilities and landscape are improved. The differences of elasticity are represented by the interaction of the dummy variable and travel cost, substitute site travel cost, and income.

The consumer surplus of participants equals the area under the expected demand function for access to Dong-Feng Cycleway. The demand in Equation (3) is semi-log. Both the choke price of current and improved lighting facilities or landscape in the demand function are infinite. When the quality of the project improves, visitors’ recreational demand shifts rightward. The change of the consumer surplus for the improvement of environmental quality can be measured as follows.
(4)$$∆CS=\frac{x^{\prime }}{\beta^{\prime }}-\frac{x}{\beta }$$
where β and β′ are the coefficient of the price variable in the demand model, *x* is the number of trips with current quality, and *x*′ is the number of trips with expected improvement of quality, respectively. 

### 3.2. Questionnaire and Sample

The questionnaire of environment quality items was designed from a number of sources and literature reviews, including Bull (2006) [4], Chen and Chen (2013) [16], and Sener et al. (2009) [21]. Cyclists’ answers to the questions in the questionnaire concerning environment quality were given on a five-point Likert scale (1 = strongly disagree, 5 = strongly agree). The survey was conducted from July to August in 2016, and 420 cyclists were asked to fill out the questionnaire. Three hundred and seventy-two respondents completed the questionnaire, yielding a response rate of 88.57%. One of the advantages of CBM is data collection. The method can reduce sample sizes from repeated observations for each individual without incurring additional costs, and it can also increase estimation efficiency [44].

## 4. Results

### 4.1. Environmental Quality of Cycleways

This study adopted exploratory factor analysis to extract the major factorial dimension of environmental quality for Dong-Feng Cycleway. Factor analysis was performed using the principal component method and the Varimax rotation procedure. There were 27 items on environmental quality in the questionnaire, and six items were dropped because their factor loading was smaller than 0.5. Five major factorial dimensions were extracted out of 21 items. Table 2 lists the results of factor analysis that show that the Eigenvalues exceed 1, explaining 62.98% of the total variance.

The first dimension of factor analysis was ‘safety’, which made up a large proportion of environmental quality and accounted for 32.16% of the variation with a reliability of 0.86. The other dimensions were ‘lighting facility’, ‘lane design’, ‘landscape’, and ‘environment cleanliness’, which accounted for a total variance of 9.26%, 8.04%, 6.81%, and 6.71%, respectively. The coefficient reliabilities for ‘lighting facility’, ‘lane design’, ‘landscape’, and ‘environment cleanliness’ were 84%, 83%, 75%, and 70%, respectively. After factor analysis, five dimensions of environmental quality were introduced into the CBM to estimate the monetary value of environment improvement for cyclists.

### 4.2. Contingent Behavior Model Estimates

This study adopted CBM, combining actual trips with intended trips to estimate the recreational benefits under the hypothetical scenarios of improved environmental quality. The improvement programs included lighting facilities and landscape, which ranked the lowest among the environment factors in the pretest and formal survey (Table 1). The lighting facilities are insufficient for cyclists to ride at night and the landscape is damaged by a soil conservation project. Factors EQF1 to EQF5 represent the factors of safety, lighting facility, lane design, landscape, and environment cleanliness, respectively. The contingent behavior model under the hypothetical scenarios includes the scenarios of improved lighting facility (model A) and landscape (model B). The definition of the variables and descriptive statistics are listed in Table 1.

The goodness-of-fit of the evaluation models are revealed by Chi-squared measure, which was calculated by likelihood ratio, and differed from 0 at the 0.01 significance level. The result indicated that the null hypothesis of all variables being equal zero was rejected. The signs of cost and substitute cost variables were consistent with the demand rule for both models and differ significantly from 0. The socioeconomic variables were positive and significantly related to participants’ age and income. Participants who are older and have higher income are more likely to ride a bike at Dong-Feng Cycleway. The older cyclists are more likely to choose bike tourism for leisure. For the perception factors, in model A (improved lighting facility), ‘lighting facility’ (EQF2), ‘lane design’ (EQF3), and ‘environment cleanliness’ (EQF5) were significantly related to cyclists’ demand. In model B (improved landscape), apart from the aforementioned three factors, ‘landscape’ (EQF4) was also found to be significantly related to the demand. Lighting facility and landscape were positively related to the demand of cyclists. When the lighting facility and landscape factors are improved, the cyclists’ intention to ride here increases. In contrast, lane design and environment cleanliness were negatively related to the demand because when the intended trips increased with in two hypothetical scenarios, the lane design and environment cleanliness factors remained constant. Thus, the relationship between the demand and lane design and environment cleanliness factors changed from positive to negative. The quality improvement dummy variables (D1, D2) were significantly different from zero at the 0.01 level, and demonstrated that the quality improvement would lead to an increase in the number of trips taken. The coefficient of the interaction between dummy variables (D1, D2) and own-price, cross-price, and income was significantly different from zero. The results showed a shift in the elasticities of the recreation demand as the environmental quality improved. The results are consistent with the research of Whitehead et al. (2000) [14]. The details of the results are listed in Table 3.

### 4.3. Elastic Estimates

The dummy variables, D1 and D2, were significantly different from 0 at the 0.01 level. Both improvement projects would lead to an increase in the number of trips taken. For the lighting facility project and the landscape project, the demand of trips increased from 3.38 to 6.11 and 6.50, respectively. The interaction coefficient between the dummy variables (D1, D2), own-price, and income was positive and significantly different from zero at the 0.01 level. However, the interaction coefficient between the dummy variable and cross-price was negative and significantly different from zero at the 0.01 level. The results are presented in Table 4. In both projects, the elasticity of own-price, cross-price, and income were smaller than 1, and the elasticity of current quality was greater than the improved quality. With the quality improvement project, price and income factors became inelastic, and demand for the cycleway rose. The result is consistent with the research of Alberini et al. (2007) [35] and Whitehead et al. (2000) [14]. 

### 4.4. Estimating Recreational Benefits and Improving Effects 

The recreational benefit was obtained from Equation (4). The average recreational benefit for a participant was NT$9796 for Model A and NT$10,133 for Model B. An increase was found in the lighting facility improvement project; the consumer surplus was raised to NT$46,444. In contrast, an improved landscape raised the consumer surplus to NT$16,188 per person. With 250,000 cyclists in 2016, the findings indicate that incremental recreational benefits could have increased to NT$ 9162.20 million if the lighting facilities were improved, and a gain of NT$1513.85 million could occur if the landscape program was better than the current situation (see Table 5).

## 5. Discussion 

The empirical results showed that the exploratory factor analysis extracted the major factorial dimensions of environmental quality for Dong-Feng Cycleway, including ‘safety’, ‘lighting facility’, ‘lane design’, ‘landscape’, and ‘environment cleanliness’. The scale of the lighting facilities and the landscape quality were the lowest among the environment factors, and became the hypothetical improvement projects in this study. The results of CBM found that improving the lighting facilities and landscape factors would increase the number of intended trips and the recreational benefits for cyclists. The average recreational benefit for a rider with the current quality of lighting and landscape is NT$9,796 and NT$10,133, respectively. After improving the quality of the lighting facilities and the landscape, the recreational benefits could be increased to NT$46,444 and NT$16,188 for cyclists.

In order to examine the validity of the quality improvement projects, this study calculated the elasticity of own-price, cross-price, and income for the current and improved quality of lighting facilities and the landscape. The results revealed that the improved quality was less elastic than the current quality. In other words, the demand for the cycleway became less elastic with the quality improvement projects. This finding is the same as the findings of Alberini et al. (2007) [35] and Whitehead et al. (2000) [14]. Cyclists would not change their decision to visit Dong-Feng Cycleway after improvement of the environmental quality.

## 6. Conclusions

This study adopted the contingent behavior method to estimate the effect of improving the environmental quality of Dong-Feng Cycleway. The theoretical model was based on the travel cost method, and the Poisson function was used in the empirical model. The respondents reported their intention to ride a bike under hypothetical scenarios of improvement of the lighting facilities and landscape. CBM, combining actual and intended behavior data, was used to measure the effect of the quality improvement projects and to calculate the recreational benefits with different scenarios of lighting facility and landscape improvement. The effect of environment quality improvement is tremendous for cyclists. According to this result, public officials or managers should to improve environmental quality of cycleways.

The estimation of elasticity proved the validity of the quality improvement effect. This paper also found that the contingent behavior method contains more information than the traditional travel cost model; the findings can assist officials to develop strategic policy concerning quality improvement to sustain bicycle tourism. 

Based on the results, this study suggests that any efforts to improve existing cycleways should not neglect the importance of lighting facilities and the surrounding landscape; and for the planning of future cycleways, efforts should be put into maximizing cyclists’ recreational benefits, and cycleway design guides should provide designers information on how to achieve that. Information of the lighting facilities and surrounding landscape should be provided to cyclists in cycleway guides.

The limitation of this study is that the samples came from on-site cyclists only. According to the structure used in the study of Whitehead et al. (2000) [14], nonparticipants should also be included in the survey. As an effect on the demand function, higher environmental quality may attract new participants to the site. To elicit more information on attracting new cyclists to use the cycleway, further research should include nonvisitors’ opinions in the survey.

## Figures and Tables

**Figure 1 ijerph-16-03460-f001:**
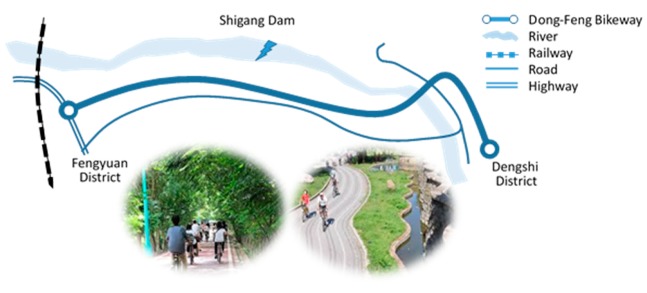
Dong-Feng Cycleway map and pictures.

**Table 1 ijerph-16-03460-t001:** Definition of the variables and descriptive statistics.

Variable	Definition	Mean	SD
TRIPS1	The number of observed trips for individual visits to Dong-Feng Cycleway under the current quality.	3.38	5.67
TRIPS2	The number of observed trips + intended trips for individual visits to Dong-Feng Cycleway under quality improvement of lighting facilities.	6.11	7.37
TRIPS3	The number of observed trips + intended trips for individual visits to Dong-Feng Cycleway under quality improvement of the landscape.	6.50	8.25
COST	Total round trip travel costs to Dong-Feng Cycleway, the cost is measured in New Taiwan dollars (NT$).	692	947
SCOST	Total round trip travel costs to a substitute site—Kenting National Park in Pingtung (NT$).	809	1,147
AGE	Cyclist age.	31.15	10.44
INCOME	The monthly income of the respondent (NT$).	25,134	19,177
EQF1	The factor score of ‘safety’. (Origin Likert scale)	- (4.44)	- (0.53)
EQF2	The factor score of ‘lighting facility’. (Origin Likert scale)	- (4.22)	- (0.68)
EQF3	The factor score of ‘lane design’. (Origin Likert scale)	- (4.23)	- (0.60)
EQF4	The factor score of ‘landscape’. (Origin Likert scale)	- (3.84)	- (0.61)
EQF5	The factor score of ‘environment cleanliness’. (Origin Likert scale)	- (4.23)	- (0.63)
D1	Dummy equal to 1 if the lighting facilities were improved in Dong-Feng Cycleway, the respondents’ intention to ride a bike there would change; 0, otherwise	0.93	0.27
D2	Dummy equal to 1 if more trees were planted to improve the landscape in Dong-Feng Cycleway, the respondents’ intention to ride bike there would change; 0, otherwise.	0.92	0.27

**Table 2 ijerph-16-03460-t002:** Factor analysis of environmental quality for cyclists.

Items	Safety	Lighting Facility	Lane Design	Landscape	Environment Cleanliness
Bicycle path pavement maintenance	0.733				
Guarantee the rights of cyclists	0.723				
Controlling steam locomotives into bicycle lanes	0.711				
Management and maintenance of public facilities around bicycle lane	0.679				
No parking for motors on bicycle lane	0.641				
Safety maintenance of the surroundings of bicycle paths	0.633				
Bicycle lane has enough lighting at night		0.805			
Bicycle lane night index visibility		0.764			
Bicycle lane night guardrail color visibility		0.757			
Bicycle lane lighting at night is bright enough		0.752			
The slope of the bicycle lane is appropriate			0.846		
Bicycle lane is properly curved			0.803		
The width of the bicycle lane is appropriate			0.795		
Bicycle lane guardrail setting			0.613		
Dispersion of landscape position				0.763	
Landscape is diversity				0.728	
The landscape is crowded				0.668	
Landscape has a famous specialty				0.657	
Appropriate location of toilets along the bicycle path					0.774
Cleanliness of use of toilets along bicycle lanes					0.770
There are enough trash bins along the bike path					0.749
Eigenvalue	6.75	1.95	1.69	1.43	1.41
Cumulative variation (%)	32.16	41.42	49.46	56.27	62.98
Cronbach’s α	0.86	0.84	0.83	0.75	0.70

**Table 3 ijerph-16-03460-t003:** Contingent behavior model for improvement effect.

Variable	Model A	Model B
Constant	0.0193 (0.382)	0.4412 (10.761)
COST	−0.0003 *** (−16.416)	−0.0003 *** (−20.732)
SCOST	0.0003 *** (15.346)	0.0003 *** (18.190)
AGE	0.0252 *** (43.977)	0.0274 *** (57.751)
INCOME	0.00007 *** (11.047)	0.00008 *** (15.777)
EQF1	0.0049 (0.585)	0.0004 (0.340)
EQF2	0.2416 *** (28.012)	0.2449 *** (31.302)
EQF3	−0.0706 *** (−9.999)	−0.0691 *** (−10.448)
EQF4	0.0099 (1.427)	0.0151 *** (2.606)
EQF5	−0.0913 *** (−13.286)	−0.0867 *** (−13.700)
D1	0.3948 *** (12.983)	-
D1 COST	0.0001 *** (4.806)	-
D1 SCOST	−0.0002 *** (−13.001)	-
D1 INCOME	0.00004 *** (3.860)	-
D2	-	1.7936 *** (5.507)
D2 COST	-	0.0004 *** (2.989)
D2 SCOST	-	−0.0002 *** (−13.353)
D2 INCOME	-	0.0006 *** (6.118)
Chi-squared	984 ***	1138 ***
Observation	784	784

Note: *** *p* < 0.01, t values in parentheses.

**Table 4 ijerph-16-03460-t004:** Elasticity estimates.

Elasticity	Lighting Facility	Landscape
Current quality elasticity		
Own-price	−0.125953	−0.121762
Cross-price	0.303048	0.273678
Income	0.512112	0.611752
Improved quality elasticity		
Own-price	0.026565	0.076216
Cross-price	−0.114006	−0.100279
Income	0.166571	0.232376

**Table 5 ijerph-16-03460-t005:** Recreational benefits and programs effect.

Value (1000 NT$)	Lighting Facility	Landscape
Recreational benefits (average)	9.80 to 46.44	10.13 to 16.19
Incremental of improvement effect	36.65	6.06
Total recreational benefits	11,611,120	4,047,072

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
