# Peer review of "The Effect of Improving Cycleway Environment on the Recreational Benefits of Bicycle Tourism"

_ijerph, 2019, doi:10.3390/ijerph16183460_

Round 1

Reviewer 1 Report

The article focusses on the evaluation of environment quality in prospect to develop bicycle tourism in Taiwan adopting the method of contingent behavior. The research is built on a widespread literature considering the main issues on bicycle tourism and bike routes. I suggest describing in a more detailed way how CBM has been used in the past and why it is deemed appropriate to estimate recreational benefits.  The research is based on a sound methodology, still the description of the methodology should be summed up in the methods section and not in the findings. More data on the sample would be useful. In the conclusions and discussion section the practical contribution (who should use these results?) and the limitations of the research should be explained.

I wish you good luck with the revision.

Author Response

Reviewer 1

We appreciate your helpful comments, and guidance to revise the manuscript. The manuscript has been revised according to your recommendations. We hope the following explanations answer all your suggestions (they are marked in blue prints in the manuscript).

We have added a discussion concerning CBM (please see the first three paragraphs of Materials and Methods section, page 3-4.). We have added two paragraphs about practical contribution and limitation in the conclusion section (please see the third and fourth paragraph in the conclusion section, page10).

Deely, J, Hynes, S, & Curtis, J (2019). Combining actual and contingent behaviour data to estimate the value of coarse fishing in Ireland. Fisheries Research,215, 53-61.

Jeon, H. & Herriges, J. A. (2017). Combining Revealed Preference Data with Stated Preference Data: A Latent Class Approach. Environmental and Resource Economics, 2017, 68 (4), 1053–1086.

Reviewer 2 Report

The main viewpoints and results of this paper are clearly illustrated, but the objective and significance of this research are not well demonstrated. Why is the “mountain bike trails” emphasized without further discussion? In the same line, it would be better to describe how the environmental requirement for bicycle tourism is different from others. This paragraph is confusing due to the imprecise and undefined concepts. A map and/or table demonstrating the location, surrounding environments, and other basic information of the Tong-Fon Bikeway are needed. The questionnaire design, sampling, and scenario setting (how are lighting facility and landscape improved) should be demonstrated in detail. A table showing the descriptive statistics of variables needs to be provided before presenting the regression results. The negative association between demand and lane design and environment cleanness does not seem reasonable and needs to be explained. The coefficients of cross terms also need further explanation and deeper discussion. There are no convincing reasons to only investigate the impacts of lighting facility and landscape (although in line 205 there is a sentence “the scale of lighting facility and landscape were the lowest” to explain, it’s not an acceptable reason). The age variation of the effects also worth discussing. In part 5 and part 6, the significance of this study should be elaborated, and there are few practical recommendations for management improvement. Line 77, the abbreviation “WTP” lacks an illustration. Line 144, the number “67.19%” is different from that in table 1 (62.98%). Besides, there are many mistakes in format, terminology, grammar, and wording.

Author Response

Reviewer 2

We really appreciate your valuable comments. Thanks to your comments, we are able to improve the quality of the manuscript. The manuscript has been revised following your suggestions. We hope the following answers all of your queries.

The objective and significance of the study have been stated in the introduction. (Please see the first and second paragraph in the introduction section, on page 1-2). A map and pictures of Dong-Feng Bikeway have been added to the introduction section (see figure 1, on page 2). Questionnaire design and sampling are discussed in 3.2 (see row 196 to 204, page 5). The hypothetical scenarios are described in 3.1 (see the third paragraph of contingent behavior method, from row 142 to 151, page 3-4), and 4.1 the sub-section of contingent behavior model estimates (see the first paragraph of contingent behavior model estimates, from row 225 to 233, on page 6). Because the lighting facility is not enough for cyclists to ride at night, and the landscape is damaged by soil conservation project. A table is added to describe the definition of the variables and descriptive statistics (see Table 2, on page 7). The negative association between demand and lane design and environment cleanness is explained at the end of 4.2. “When lighting facility and landscape factors are improved, cyclists’ intention to ride here increases. In contrast, lane design and environment cleanness were negatively related to the demand because when the intended trips increased with the two hypothetical scenarios, lane design and environment cleanness factors remained constant. Thus the relationship between the demand and lane design and environment cleanness factors changed from positive to negative.” (see from row 245 to 250, on page 6) The quality improvement dummy variables (D1, D2) were significantly different from zero at the 0.01 level; this demonstrated that the quality improvement would lead to an increase in the number of trips taken. The coefficient on the interaction between dummy variables (D1, D2) and own-price, cross-price, and income was significantly different from zero. The results showed a shift as well as a change in elasticities of the recreation demand as the environmental quality improved. The results are consistent with the research of Whitehead et al. (2000) [14]. The details of the results are listed in Table 3 (see from row 250 to 257, on page 6-7). The socioeconomic variables were positive and significantly related to participants’ age and income. Participants who are older and have higher income are more likely to ride a bike at Dong-Feng Bikeway. The older cyclists enjoy cycling for leisure (see from row 237 to 240, on page 6). The abbreviation “WTP” - willingness to pay is provided in row 117, page 3, and the total variance is corrected to 62.98% (see row 213, on page 5).

Thank you very much again; we look forward to hearing from you soon.

Best regards,

Round 2

Reviewer 2 Report

The authors have addressed all concerns of mine. I have no further comments/suggestions.